# Polysaccharides Produced by Plant Growth-Promoting Rhizobacteria Strain *Burkholderia* sp. BK01 Enhance Salt Stress Tolerance to *Arabidopsis thaliana*

**DOI:** 10.3390/polym16010145

**Published:** 2024-01-03

**Authors:** Enni Chen, Changsheng Yang, Weiyi Tao, Shuang Li

**Affiliations:** 1College of Biotechnology and Pharmaceutical Engineering, Nanjing Tech University, Nanjing 211816, China; 202161118021@njtech.edu.cn (E.C.); 202261218122@njtech.edu.cn (C.Y.); 2College of Food Science and Light Industry, Nanjing Tech University, Nanjing 211816, China; taoweiyi@njtech.edu.cn

**Keywords:** polysaccharide, *Burkholderia* sp., *Arabidopsis thaliana*, biostimulants, salt stress, PGPR

## Abstract

Salt stress is one of the most serious abiotic stresses leading to reduced agricultural productivity. Polysaccharides from seaweed have been used as biostimulants to promote crop growth and improve plant resistance to abiotic stress. In this study, PGPR strain *Burkholderia* sp. BK01 was isolated from the rhizosphere of wheat, and it was characterized for phosphorus (Pi) dissolution, indole-3-acetic acid (IAA) production, ammonia (NH_3_) and exopolysaccharides (EPS). In particular, strain BK01 can efficiently produce extracellular polysaccharide with a yield of 12.86 g/L, using sorbitol as carbon source. BK01 EPS was identified as an heteropolysaccharide with Mw 3.559 × 10^6^ Da, composed of (D)-galactose (75.3%), (D)-glucose (5.5%), (L)-rhamnose (5.5%), (D)-galactouronic acid (4.9%) and (D)-glucuronic acid (8.8%). The present work aims to highlight the effect of the BK01 EPS on growth and biochemical changes in *Arabidopsis thaliana* under salt stress (100 mM). The purified BK01 EPS at a concentration of 100 mg/L efficiently promoted the growth of plants in pot assays, improved the chlorophyll content, enhanced the activities of SOD, POD and CAT, and decreased the content of MDA. This results suggested that the polysaccharides produced by PGPR strain *Burkholderia* sp. BK01 can be used as biostimulants to promote plant growth and improve plant resistance to salt stress.

## 1. Introduction

Soil salinization can be caused by many things, including saline irrigation, water scarcity, rising sea levels due to global warming, and the large-scale application of compost fertilizers [1]. Currently, soil salinization has resulted in lower seed germination rates, slower plant growth, and reduced crop yields worldwide [2]. To mitigate the effects of soil salinity on agriculture production, numerous strategies, such as developing salt-tolerant crops by breeding or plant genetic engineering, and various agricultural practices including the applications of plant growth-promoting rhizobacteria (PGPR) and seed-biopriming techniques have been applied to improve the salt tolerance of plants, stimulate plant growth, and increase production under salt stress [1,3].

PGPR microorganisms including species in the genus of *Rhizobium* [4], *Pseudomonas* [5], *Pantoea* [6], *Paenibacillus* [7], *Burkholderia* [8], *Kosakonia* [3], *Bacillus* [3], *Achromobacter* [3], *Ochrobactrum* [9] have been reported to increase the tolerance of plants to salt and other abiotic stresses. The genus *Burkholderia* is phylogenetically diverse and consists of multiple deep-branching 16S rRNA lineages. Recent studies have shown that with the exception of a few strains, *Burkholderia* sp. are pathogenic. Most species of the genus *Burkholderia* have many beneficial functions for plants and the environment, such as bioremediation, nitrogen fixation, bioprophylaxis, induction of plant resistance, etc. [10]. Therefore, *Burkholderia* has become a hot research topic of PGPR, as a microbial resource with important applications [11]. However, most PGPR are non-sporulate bacteria, such as *Burkholderia*, *Pseudomonas*, *Pantoea* and *Kosakonia*, which often limit their commercial development as microbial agents due to their inability to form spore-like hypopus for long-term storage. Based on the secretions of the PGPR, some widely recognized modes of action that assist plants in salt tolerance have been proposed. For instance, the auxin indoleacetic acid (IAA) is a kind of plant growth hormone, and PGPR strains can usually synthesize a certain amount of IAA to promote the growth of host plants and enhance salt tolerance [12]. Antioxidant enzymes (AEs), such as catalase, are also a key probiotic product of PGPR strains for their host plants; they play an important role in promoting the accumulation of abscisic acid and degrading reactive oxygen species (ROS) in plants, thereby limiting the oxidative damage caused by salt stress [13].

Polysaccharides are carbohydrates derived from microorganisms, plants, or animals. Due to their significant structural features, physical and chemical properties, and biological activities, polysaccharides are widely applied in foods, medicines, polymer materials, cosmetics, and agriculture [14,15]. In particular, some exopolysaccharides (EPSs) produced by PGPR have been shown to play an important role in combating combat harsh environmental conditions like salinity and drought [16]. Although it has been noted that some PGPR strains can over-produce EPSs, the roles of EPSs produced by PGPR bacteria in abiotic stress alleviation during plant growth under the stress of salt are still unclear. Compared with the well-known application of seaweed polysaccharides in promoting plant growth and stress resistance, EPSs from PGPR strains remain to be further developed.

Here, we report for the first time on the novel exopolysaccharides produced by PGPR strain *Burkholderia* sp. BK01 could enhance salt stress tolerance to *Arabidopsis thaliana.* The BK01 EPSs, as biostimulants, are easy to prepare industrially, due to their high yield, easy extraction, and long-term storage. This work provides a new insights to a better application of PGPR by developing EPS products from PGPR.

## 2. Materials and Methods

### 2.1. Isolation and Identification of Strain BK01

*Burkholderia* sp. BK01 (Strain BK01) was isolated from the roots of wheat grown in saline soil in Dongying, China. Genomic DNA of strain BK01 was prepared using a genomic DNA purification kit. The general bacterial primers 27F (5′-AGAGTTTGATCMTGGCTCAG-3′) and 1492R (5′-TACGGYTACCTTGTTACGACTT-3′) were used to amplify the 16S rDNA gene. The PCR products were purified and sequenced with GenScript (Nanjing, China), and the obtained sequences were analyzed using the nucleotide module of the Basic Local Alignment Search Tool (BLASTN, Bethesda, MD, USA) on the National Center for Biotechnology Information (NCBI) website. Strain BK01 was recovered from storage at −80 °C by subculture on LB agar plate and subsequently grown on LB agar plate containing 2% (*w*/*v*) sorbitol at 30 °C for 5 days to observe the colony morphology.

### 2.2. Plant Growth-Promoting Substances of Strain BK01

#### 2.2.1. Phosphate Solubilization

The defined medium NBRIP [17] broth was used for a quantitative assay of the phosphate solubilization activity of strain BK01. Briefly, 25 mL of NBRIP medium (10 g/L glucose, 5 g/L Ca_3_(PO_4_)_2_, 5 g/L MgCl_2_·6H_2_O, 0.25 g/L MgSO_4_·H_2_O, 0.2 g/L KCl, 0.1 g/L (NH_4_)_2_SO_4_, pH 7) was inoculated with 1 mL of bacterial suspension (OD_600_ = 16.29) and incubated at 30 °C for 5 days. Phosphate in the culture supernatant was estimated using the Fiske and Subbarow method [18]. The data were means of three experiments.

#### 2.2.2. Production of Indole-3-Acetic Acid (IAA)

The IAA produced by strain BK01 was quantitatively assessed using the modified method of Majeed et al. [19]. For the assay, 1 mL of BK01 suspension (OD_600_ = 16.29) was added to 25 mL of LB broth containing 100 mg/L of L-tryptophan and incubated for 5 days at 30 °C. IAA concentration in the culture supernatant was estimated using Salkowski reagent (2% 0.5 FeCl_3_ in 35% HClO_4_ solution) [20]. The data were means of three experiments.

#### 2.2.3. Production of Ammonia

The ammonia production capacity was determined qualitatively by the improved method of Rana et al. [21]. The test strains were inoculated in peptone broth (peptone 10 g/L; NaCl 5 g/L) and incubated at 30 °C for 48 h. Detection of ammonia production was carried out by adding 500 μL Nessler’s reagent to the 48 h old culture. Development of a brown to yellow color indicated the presence of ammonia.

### 2.3. Characterization of BK01 EPS

#### 2.3.1. Preparation of BK01 Exopolysaccharide (BKEPS)

The genus *Burkholderia* has the ability to produce several types of extracellular polysaccharides (EPSs). Bartholdson et al. [22] reported that sugar alcohol induced exopolysaccharide biosynthesis in the *Burkholderia* genus, in particular, mannitol and sorbitol. In order to obtain optimal carbon sources for EPS production in strain BK01, mannose, fructose, glycerol, sorbitol, sucrose, and glucose were used as carbon sources.

Strain BK01 was cultured in seed medium (20 g/L of carbon source, 2 g/L of yeast extract, 2 g/L of K_2_HPO_4_·3H_2_O, 0.1 g/L of MgSO_4_·7H_2_O, at a pH of 7.0–7.2) at 30 °C and 200 rpm for 24 h. Then, the seed culture (6%, *v/v*) was inoculated into fermentation medium (40 g/L of carbon source, 5 g/L of yeast extract, 2 g/L of K_2_HPO_4_·3H_2_O, 0.1 g/L of MgSO_4_·7H_2_O), and the fermentation was carried out at 30 °C and 200 rpm for 72 h.

Viscosities of fermentation broth were measured at 30 °C with a rotary viscometer (IKAROTAVISC lo-vi, Staufen, Germany) and spindle 3 at 30 rpm. The BK01 EPS in fermentation broth was extracted and purified by alcohol precipitation with ethanol (1:3, *v*/*v*), deproteinization using Sevage reagent (CHCl_3_:BuOH = 5:1, *v*/*v*), dialysis, and lyophilization, following the method of Huang et al. [23].

#### 2.3.2. Chemical and Monosaccharide Composition Analysis of BK01 EPS

The purified BK01 EPS was redissolved in deionized water at room temperature and a concentration of 5 g/L. The samples were then stored for at least 12 h to ensure their full hydration. Total sugar and protein contents in BK01 EPS were quantified by the H_2_SO_4_-phenol method [24] and the Bradford method [25], respectively.

The hydrolysis of BK01 EPS for monosaccharide composition analysis was carried out according to the methods provided by Huang et al. [23]. The hydrolysis supernatant was characterized by IC (ionic chromatography) analysis. The sample was loaded on an ICS5000 system (ThermoFisher, Waltham, MA, USA) equipped with a DionexCarbopac TM PA20 column at 30 °C. A mixture of 16 standard monosaccharides was prepared (fucose, rhamnose, arabinose, galactose, glucose, xylose, mannose, fructose, ribose, galacturonic acid, glucuronic acid, aminogalactose hydrochloride, glucosamine hydrochloride, N-acetyl-D-glucosamine, guluronic acid and mannuronic acid) (Maikelin Biology Science and Technology Co., Ltd., Shanghai, China), and each monosaccharide standard solution was precisely configured as standard. According to the peak area of the standard and the sample, the concentration of the sample was calculated, and the monosaccharide molar ratio of the sample was obtained according to the mass of different monosaccharides.

#### 2.3.3. Determination of Molecular Weight of BK01 EPS

The molecular weight of BK01 EPS was analyzed using Waters 2695 HPLC (with a 2410 oscillometric refractive detector and Empower workstation). The column used was an Ultrahydrogel™ linear gel filtration column (7.8 × 300 mm) (Waters, Milford, MA, USA). The detection conditions were 40 °C, 0.5 mL/min, and the mobile phase was 0.1 M sodium nitrate solution.

### 2.4. Plant Growth and NaCl Treatment

*Arabidopsis* seeds (*Arabidopsis thaliana* wild-type Col-0) for the test were provided by Beijing Shengshihuifeng Technology Co., Ltd. (Beijing, China). Sterilized seeds were synchronized sown in the pots with nutrition soil. Plants were maintained in a 25/19 °C day/night temperature cycle with a 16/8 h light/dark cycle. After the third leaf of *Arabidopsis* seedlings were fully developed, plants were transferred to individual pots containing nutrition soil and randomly divided into four groups of 20 plants each. To define whether the BKEPS had an effect on the growth of *Arabidopsis* in normal and saline conditions, four treatment combinations were prepared as follows: CK/0 mM NaCl, salt/100 mM NaCl, BKEPS 100 ppm/100 mM NaCl, and BKEPS 100 ppm/0 mM NaCl. The newly transplanted *Arabidopsis* seedlings in pots with or without BKEPS at 100 ppm were cultured for 7 days, and then either 0 or 100 mM of NaCl was added to the pots and the seedlings were further cultured for 7 days. After the total two weeks of growth, plants were randomly chosen and harvested to measure their fresh weight (FW), shoot length, root length, and relative chlorophyll content (SPAD-502Plus Chlorophyll Content Tester, KONICA MINOLTA, Tokyo, Japan); leaves were sampled and stored at −80 °C for further bio-chemical analysis.

### 2.5. Measurement of Biochemical Indicators

#### 2.5.1. K^+^ /Na^+^ Concentrations

Na^+^ and K^+^ concentrations were determined according to the method [26]. The leaf samples were dried at 60 °C overnight. Samples of 0.5 g were placed in a Muffle furnace and incinerated for 6 h, and then the ashes were dissolved in 5 mL of concentrated nitric acid with 500 mL of distilled water. The ion concentrations were measured using a atomic absorption spectrometer (Perkin Elmer 900 T, Waltham, MA, USA).

#### 2.5.2. Malondialdehyde (MDA) Concentration

MDA content in plants were determined using the thiobarbituric acid (TBA) reaction, following the method of Buono et al. [27]. The leaf samples (0.5 g) were homogenized in 10% (*w*/*v*) TCA, and then the supernatant of homogenate (2 mL) was mixed with 2 mL of 0.6% (*w*/*v*) TBA. The mixture was heated in boiling water for 15 min and cooled down to room temperature, then centrifuged at 10,000× *g* for 10 min. The absorbance of the supernatant was read at 450, 532, and 600 nm. The MDA contents were recorded as μmole·mg^−1^ fw.

#### 2.5.3. Proline Concentration

The proline content was measured in μg·g^−1^ fw by the method of Bates [28] with some modifications. Leaf samples (0.5 g, fw) were cut into pieces and placed in test tubes with the addition of 5 mL of 3% sulfosalicylic acid. The tubes were placed in boiling water for 10 min, and 2 mL of the supernatant was taken to mix with 2 mL of acetic acid and 3 mL of 2.5% ninhydrin. The mixture was placed in boiling water for 40 min, and then was extracted with 4 mL of methylbenzene. The absorbance of the organic phase was measured at 520 nm. The proline content was calculated according to the proline standard curve.

#### 2.5.4. Determination of Antioxidant Enzymes SOD, CAT and POD

Leaf samples (0.5 g, fw) were homogenized in liquid nitrogen and dissolved in 5 mL of 0.2 M cold sodium phosphate-buffered solution (pH 7.8). Then, homogenates were centrifuged at 12,000× *g* and 4 °C for 15 min. The fresh supernatants were immediately used to determine enzyme activities. The Bradford method was used to determine total soluble protein [25].

Superoxide dismutase (SOD) activity was assayed by the photochemical reduction of b-nitro blue tetrazolium chloride (NBT), and SOD activity was expressed as units mg^–1^ protein [29]. Catalase (CAT) activity was assayed based on the decreasing of H_2_O_2_ contents as measured at 240 nm [30]. Peroxidase activity (POD) was assayed by the oxidization of guaiacol and was calculated from the formation of the guaiacol dehydrogenation product per minute, defined as nmol min^−1^ mg^−1^ protein [31].

### 2.6. Statistical Assessment

The error bars indicate the standard deviations and the data represent the arithmetical averages of three replicates; a one-way analysis of variance (ANOVA) and least significant differences (LSD) test (*p* < 0.05) were used to calculate the standard deviations. Statistical analysis was performed using the social science statistical package Origin 2021 (Origin, Redwood City, CA, USA).

## 3. Results and Discussion

### 3.1. Identification of Burkholderia sp. BK01

The strain BK01 is a Gram-negative, rod-shaped bacterium, with cells measuring approximately 1–1.5 μm in length and 0.3–0.5 μm in width (Figure 1A). The colonial morphology of BK01 exhibited a moist, smooth surface and secreted a large amount of viscous substance when grown in sorbitol medium (Figure 1B). BLAST analysis revealed that the 16S rDNA sequence of BK01 (GenBank accession number: OR656448) showed the highest identity (99%) with *Burkholderia gladioli* (Figure 1C). Therefore, strain BK01 was named *Burkholderia* sp. BK01, and has been deposited at the China Center for Type Culture Collection (CCTCC NO: M 20231319). The genus *Burkholderia* comprises more than 60 species isolated from a wide range of niches, and among them, more than 30 non-pathogenic species have been found to be associated with plants and considered to be potentially beneficial. The environmental and potentially beneficial plant-associated Burkholderia share characteristics such as a quorum sensing system, the presence of nitrogen fixation and/or nodulation genes, and the ability to degrade aromatic compounds [32]. PGPR strains such as *Burkholderia phytofirmans* PsJN [8] and *Burkholderia pyrrocinia* CNUC9 [33] were reported to enhance salt tolerance in *Arabidopsis thaliana*.

### 3.2. Plant Growth-Regulating Substances Produced by Strain BK01

As PGPR, strain BK01 exhibited an excellent ability to produce plant growth-regulating substances such as Pi-solubilization, IAA and NH_3_ production, shown in Figure 2. 

For the Pi solubilization test, the pH of culture decreased from the initial 7.47 to 5.67, and the available dissolved phosphorus reached a high of 373.5 μg/mL on the fifth day (Figure 2A). This mean strain BK01 could reduce the pH by secreting acid, and then dissolve insoluble inorganic phosphorus into bioavailable soluble phosphorus.

IAA is a widely studied physiologically active growth hormone that can be synthesized by PGPR, using tryptophan as a precursor and through different pathways [34]. The strain BK01 could produce the maximum IAA 29.56 mg/L at 100 mg/L tryptophan after 4 days incubation (Figure 2B).

Some PGPR strains have ammonia transporter proteins in their cells, which are thought to be involved in the cellular reabsorption of NH_4_^+^ [35]. Taking the two randomly selected *Burkholderia* strains as controls, strain BK01 exhibited a higher ammonia production capacity, with a darker yellow color reaction (Figure 2C).

### 3.3. Production and Characterization of EPSs

*Burkholderia* species are able to produce several types of exopolysaccharides (EPSs). EPS production by *Burkholderia* is tightly regulated as a response to external conditions, such as carbon source and growth condition. It has been reported that plant host and sugar alcohol induce exopolysaccharide biosynthesis, in particular mannitol and sorbitol [22]. Six kinds of carbon source (mannose, fructose, glycerol, sorbitol, sucrose and glucose) were tested for the cell growth and EPS production of strain BK01, as shown in Figure 3: All six carbon sources can be used for growth, but only mannitol, fructose and sorbitol can be used to over-produce EPSs. Among them, sorbitol was the optimum carbon source for EPS production; the viscosity of the thick broth reached 2100 mPa·s, and the EPS yield was 12.86 g/L. Although the structure of polysaccharides from *Burkholderia* species has been summarized in the literature [36,37], there has been little data reported on their yields. At present, the EPS productivity of high-yield strains has excellent industrialization prospects for sphingans (such as welan, gellan, sanxan and rhamsan), being about 20–30 g/L [38]. Therefore, strain BK01 can be considered as a high-yield strain with industrialization prospects, and its polysaccharide yield could be further improved through the optimization of medium components and the fermentation process.

The BK01 EPS was extracted and purified by alcohol precipitation, deproteinization, dialysis and lyophilization. The purified BK01 EPS was obtained as off-white fluffy powder, and it was named BKEPS. The total carbohydrate content of BKEPS was 61.25%, the uronic acid content was 30%, and the protein content was 10.1%. The monosaccharide composition of BKEPS was determined by HPLC, revealing (D)-galactose, (D)-glucuronic acid, (D)-glucose, (L)-rhamnose, and (D)-galactouronic acid as the major components, as shown in Table 1. The content of up to 75.3% (D)-galactose and 4.9% (D)-galacturonic acid make this newly obtained BKEPS very recognizable. The exopolysaccharides synthesized by the genus *Burkholderia* have rich structural diversity, and at least seven different EPSs have been identified, and their structure determined [39]. Cepacian is the major EPS produced by *Burkholderia*; it is composed of a branched acetylated heptasaccharide repeat-unit with (D)-glucose, (L)-rhamnose, (D)-mannose, (D)-galactose, and (D)-glucuronic acid in a ratio of 1:1:1:3:1 [40]. In terms of composition, BKEPS is significantly different from Cepacian. However, the structure of BKEPS remains to be further analyzed.

The weight average molar weight (Mw) and number average molar weight (Mn) values of BKEPS were 3.559 × 10^6^ Da and 1.835 × 10^5^ Da, respectively. The polydispersity (Mw/Mn) value was up to 19.4, which suggested that the molecular weight distribution of BKEPS was relatively broad [41]. Therefore, BK01 polysaccharides with different chain lengths were dispersed in aqueous solution, rather than particles with uniform size.

### 3.4. Functional Characterization of BKEPS 

Seaweed polysaccharides can be used as biostimulants to promote crop growth and improve plant resistance to abiotic stress. Zuo et al. reported that low-molecular-weight polysaccharides from seaweed (*Ulva prolifera*) in concentrations of 0.01%, 0.03% and 0.05% could enhance the tolerance of wheat (*Triticum aestivum*) to osmotic stress; the fresh weights and shoot lengths of seedlings treated with polysaccharide were significantly increased. Also, the activities of antioxidant enzymes were enhanced, and the malondialdehyde content reduced [42]. However, compared with seaweed polysaccharides, microbial polysaccharides are easier to industrialize, as they are very quick to produce under fully controlled fermentation conditions. Exopolysaccharides are the main components of microbial extracellular of microorganisms, and so are widely present in PGPRs, affecting the growth of plants in many ways. Exopolysaccharides released by rhizobia play a pivotal role in both establishment of effective symbiosis with leguminous plants and adaptation to environmental stresses [43].

Recently, the exopolysaccharides synthesized by PGPR strains have been noted to play an important role in enhancing salt tolerance to plants. The exopolysaccharides produced by PGPR strain *Pantoea alhagi* NX-11 affected the ability of the strain to enhance the salt tolerance of rice seedlings [6]. Another example is halo-PGPR *Halomonas* sp. Exo1; its purified EPS may find applications as bioinoculants or biofertilizers in the cultivation of salt-tolerant crops such as rice in low-lying contaminated coastal areas [44].

To examine the influence of BKEPS polysaccharide on the salt tolerance of plants, week-old *Arabidopsis* plants were grown in nutrient soil containing 0 and 100 ppm BKEPS with or without 100 mM NaCl. After salt stress for 7 days, the seedlings were photographed, and their FW and root/shoot length were measured. Differences were observed in the phenotypes of plants in the four treatment groups under both normal and salt stress conditions (Figure 4). The growth of seedlings was depressed under salt stress induced by 100 mM NaCl (group B); Exogenous application of BKEPS polysaccharide could significantly improve plant growth performance, both under normal condition (group D) and under salt stress (group C). After 7 days in the salt stress, *Arabidopsis* seedlings in the group B (100 mM NaCl) exhibited reduced growth and increased leaf chlorosis; however, plants with BKEPS 100 ppm treatment in group C (BKEPS + NaCl) grew better than those in group B, exhibiting a significant 170% increase in FW, a 39.6% greater root length, a 79.2% greater shoot length, and a 110% higher relative chlorophyll content (Table 2). It was particularly noteworthy that exogenous application of BKEPS polysaccharide also showed significant growth promotion performance under normal conditions. *Arabidopsis thaliana* in group D (BKEPS 100 ppm) grew better leaves than those in group A (0 mM NaCl), exhibiting a 147% higher FW, a 9.2% greater root length, a 31.2% greater shoot length, and a 12.7% higher chlorophyll content (Table 2). Together, these findings indicated that BKEPS had the potential to be used as biostimulants for the growth of plants under saline stress.

#### 3.4.1. BKEPS Improved the K^+^/Na^+^ Ratio of Arabidopsis Seedlings under Salt Stress

Many studies have shown that PGPR strains can augment the growth of plant by altering the selectivity of ions and preserving the higher K^+^ concentration relative to Na^+^ [1]. A high K^+^/Na^+^ ratio is one of the indicators of effective mechanisms to defend against salt stress [45]. Salt treatment rapidly decreased the K^+^/Na^+^ ratio in leaves after 7 days of salt treatment; the K^+^/Na^+^ ratio in group B (NaCl) was approximately 3.21 times lower than those in group A (CK). However, the K^+^/Na^+^ ratio in group C (BKEPS + NaCl) was 38.2% higher than those in group B. Plants in group D (BKEPS) had a higher K^+^/Na^+^ ratio in the leaves than those in group A under normal conditions (Figure 5). This suggested that BKEPS at a concentration of 100 ppm could help alleviate the occurrence of osmotic stress and ionic toxicity in plants and reduce the intensity of salt stress injury to plants, thus promoting the growth of *Arabidopsis* seedlings.

#### 3.4.2. BKEPS Inhibits MDA Production and Enhances Antioxidant Enzyme Activities

Plants will overproduce reactive oxygen species (ROS) under salt stress, resulting in oxidative damage. Malondialdehyde (MDA) is mainly derived from the peroxidation of membrane lipid, which is closely related to abiotic stress in plants [46]. As shown in Figure 6A, salt stress has a significant impact on the MDA content of plants; there was no significant difference between group A (CK) and group D (BKEPS), but both groups were significantly lower than group B (NaCl) and group C (BKEPS + NaCl). More specifically, MDA in group B (NaCl) was almost six-fold higher than that in group A (CK). However, the MDA content of plants treated with exogenous 100 ppm BKEPS for 7 days decreased significantly under salt stress. MDA in group C (BKEPS + NaCl) was 53.3% lower than that in group B (NaCl).

Catalase (CAT), superoxide dismutase (SOD), and peroxidase (POD) are well-known antioxidant enzymes for eliminating peroxides in living organisms. As shown in Figure 6B–D, the activities of CAT, SOD, and POD exhibited similar trends in the four groups. The lowest activity of the three enzymes was found in the group A (CK); however, the highest activity of various enzymes appeared in group C (BKEPS + NaCl). Compared with the CK group (0 mM NaCl), the activities of all three enzymes increased significantly under salt stress in group B (100 mM NaCl). Notably, the activities of all three enzymes in group C (BKEPS + NaCl) were further significantly increased by 37.1%, 42.7%, and 71.4%, respectively, which suggested the exogenous 100 ppm BKEPS significantly increased the antioxidant capacity of *Arabidopsis* seedlings under salt stress.

#### 3.4.3. BKEPS Increases the Accumulation of Proline in Plants under Salt Stress

Osmotic stress produced by salt stress can cause cellular dehydration. Plants usually actively accumulate small molecules such as proline, monosaccharides and free amino acids to regulate osmotic stress [47]. As shown in Figure 7, the trend in the proline content of all groups was similar to the performance of antioxidant enzymes in Figure 6. Plants in the CK group had the lowest proline content; the proline content increased greatly under salt stress. Once again, the proline content in the BKEPS + NaCl group was 28.8% higher than that in the NaCl group. The results showed that exogenous 100 ppm BKEPS treatment could increase proline accumulation in *Arabidopsis* seedlings under salt stress.

## 4. Conclusions

In this study, PGPR strain *Burkholderia* sp. BK01 showed a high capacity for the production of EPS with a yield of 12.86 g/L. BK01 EPS is a novel polysaccharide due to its distinctive monosaccharide composition that has great potential for industrial production and development. Many studies have shown that PGPRs produce viscous extracellular secretions, which can promote plant growth and resist environmental stress. Similarly, our results proved that the BK01 EPS can promote plant growth and exert beneficial effects on salt stress response in *Arabidopsis* seedlings. Together, these results showed that BK01 EPSs could effectively increase the K^+^/Na^+^ ratio and remove ROS by increasing proline content, enhancing SOD, POD and CAT activities, and reducing MDA content, thus alleviating the damage caused by salt stress in *Arabidopsis thaliana*. In addition, this is the first report highlighting that EPSs produced by the *Burkholderia* genus can promote growth and ameliorate salt stress in *A. thaliana*. However, further works are needed to support the application of BKEPS as a biostimulant in crop plants.

## Figures and Tables

**Figure 1 polymers-16-00145-f001:**
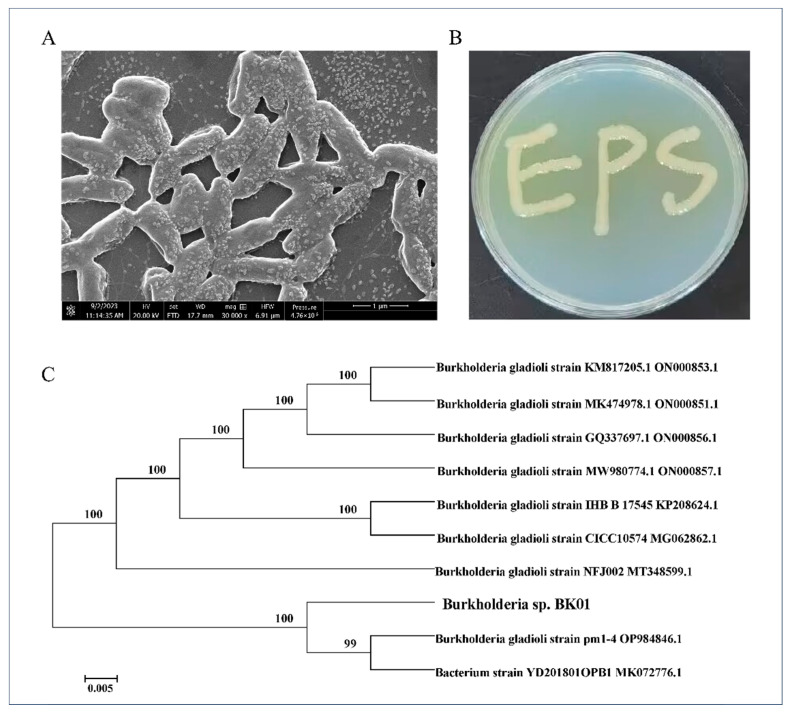
Morphological characteristics and identification of *Burkholderia* sp. BK01. (**A**) Scanning electron micrograph of the strain BK01. (**B**) The colony morphology of the strain BK01. (**C**) Phylogenetic analysis of the isolate BK01 based on the sequencing of the 16S rDNA. The scale bar indicates 0.005 substitutions per nucleotide position.

**Figure 2 polymers-16-00145-f002:**
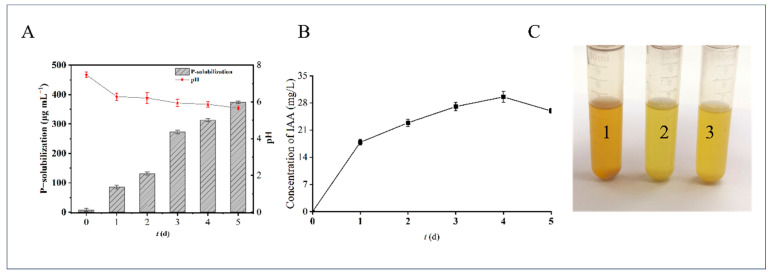
(**A**) Pi solubilization and pH in broth. (**B**) Yields of IAA produced by BK01. (**C**) Ammonia production tests of *Burkholderia* strains (1: *Burkholderia* sp. BK01; 2: *Burkholderia cepacia* ATCC 25416; 3. *Burkholderia* sp. CICC 24715). Values are three replicates.

**Figure 3 polymers-16-00145-f003:**
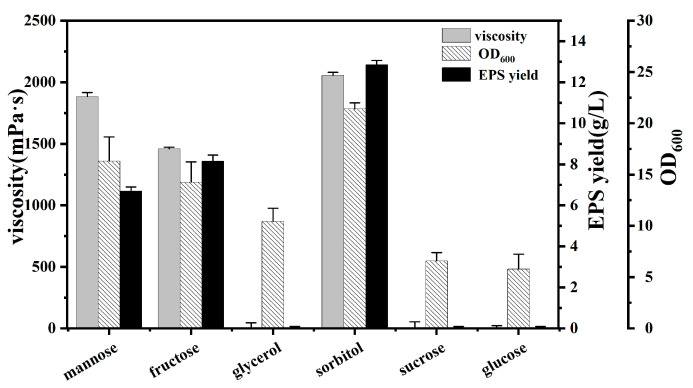
Yield of EPS, viscosity of fermentation broth, and biomass (OD_600_). Values are three replicates.

**Figure 4 polymers-16-00145-f004:**
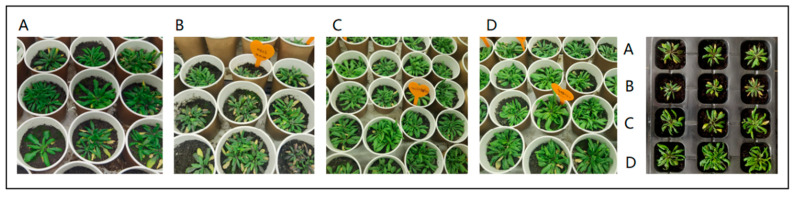
Phenotypes of *Arabidopsis thaliana* after 7 days of NaCl treatment ((**A**), control; (**B**), 100 mM NaCl; (**C**), BKEPS 100 ppm + 100 mM NaCl; (**D**), BKEPS 100 ppm); on the far right is a seedling tray with uniform specifications.

**Figure 5 polymers-16-00145-f005:**
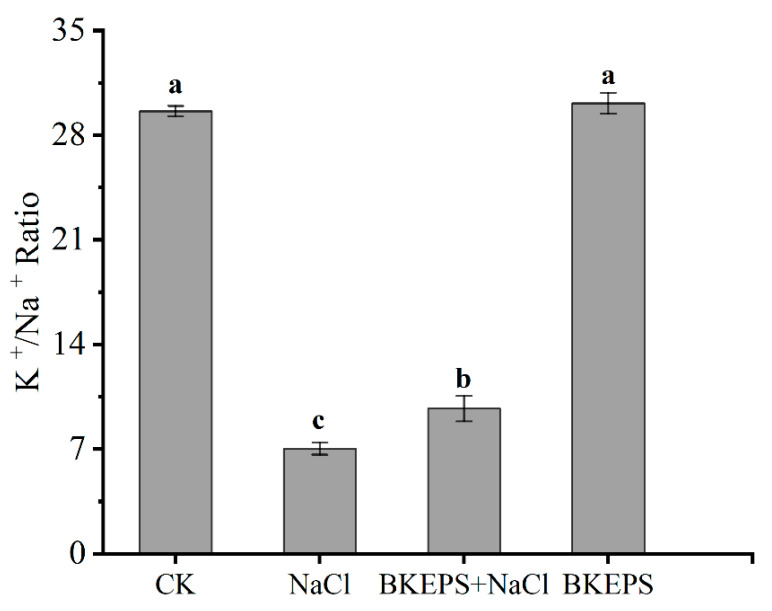
K^+^/Na^+^ ratios in *Arabidopsis* seedlings under normal (CK and BKEPS) and salt stress (NaCl and BKEPS + NaCl) conditions. Values are the mean ± LSD of three replicates. Different letters indicate significant differences at *p* < 0.05.

**Figure 6 polymers-16-00145-f006:**
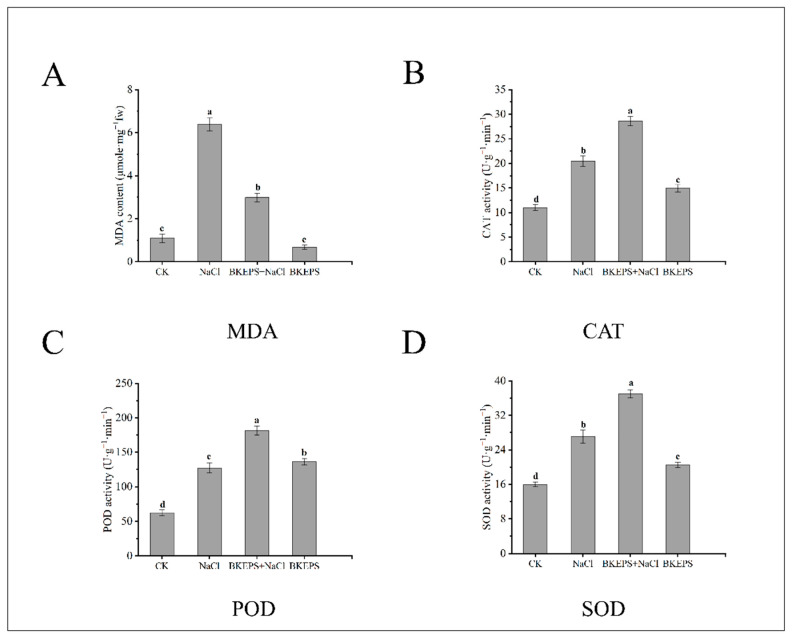
MDA contents (**A**), and CAT (**B**), POD (**C**), SOD (**D**) activities in *Arabidopsis* seedlings. Values are the mean ± LSD of three replicates. Different letters indicate significant differences at *p* < 0.05. MDA, malondialdehyde; SOD, superoxide dismutase; POD, peroxidase; CAT, catalase.

**Figure 7 polymers-16-00145-f007:**
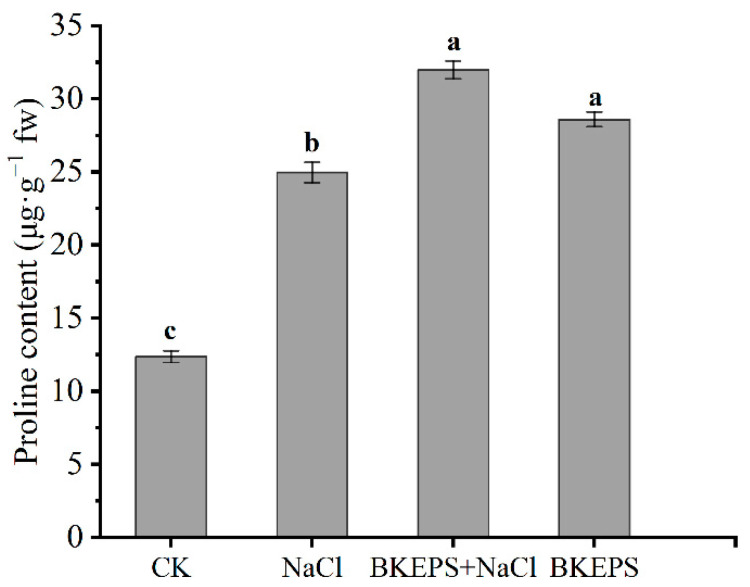
Proline contents in *Arabidopsis* seedlings. Values are the mean ± LSD of three replicates. Different letters indicate significant differences at *p* < 0.05.

**Table 1 polymers-16-00145-t001:** Preliminary characterization of BKEPS.

Parameters	BKEPS
Total carbohydrate	61.25%
Protein	10.1%
Uronic acid	30%
Molar mass moments (g/mol)	
Mw ^a^	3.559 × 10^6^
Mn ^b^	1.835 × 10^5^
PDc index	19.4
Monosaccharide composition (%)	
(L)-rhamnose	5.5
(D)-galactose	75.3
(D)-glucose	5.5
(D)-galactouronic acid	4.9
(D)-glucuronic acid	8.8

^a^ Mw and ^b^ Mn are weight-average molecular weight and number-average molecular weight, respectively.

**Table 2 polymers-16-00145-t002:** Effects of BKEPS on growth parameters of *Arabidopsis* seedlings. Values are the means ± LSD of 30 replicates. Different letters indicate significant differences at *p* < 0.05.

Groups	Treatment	Fresh Weight (mg)	Root Length (cm)	Shoot Length (cm)	Chlorophyll Relative Content (SPAD)
A	CK/0 mM NaCl	340 ± 50 c	6.73 ± 0.92 a	0.53 ± 0.10 b	38.4 ± 2.5 b
B	Salt/100 mM NaCl	226 ± 60 d	5.21 ± 0.43 b	0.24 ± 0.05 d	17.6 ± 2.0 c
C	BKEPS 100 ppm/100 mM NaCl	619 ± 50 b	6.90 ± 0.35 a	0.41 ± 0.03 c	37.4 ± 1.7 b
D	BKEPS 100 ppm/0 mM NaCl	935 ± 60 a	7.61 ± 0.16 a	0.66 ± 0.06 a	47.1 ± 2.9 a

## Data Availability

Data are contained within the article.

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
