# Peer review of "Polysaccharides Produced by Plant Growth-Promoting Rhizobacteria Strain Burkholderia sp. BK01 Enhance Salt Stress Tolerance to Arabidopsis thaliana"

_polymers, 2024, doi:10.3390/polym16010145_

Round 1

Reviewer 1 Report

Comments and Suggestions for Authors

The study "Polysaccharides produced by PGPR strain Burkholderia sp. BK01 enhance salt stress tolerance to Arabidopsis thaliana" investigated the effects of a PGPR strain Burkholderia sp. BK01, which was isolated from the rhizosphere of wheat and characterized for its ability to dissolve phosphorus (Pi), produce indole-3-acetic acid (IAA), ammonia (NH3), and exopolysaccharides (EPS). The findings suggested that polysaccharides produced by PGPR strain Burkholderia sp. BK01 could be used as bio-stimulants to promote plant growth and improve plant resistance to salt stress. However, the manuscript required further revisions to improve the clarity and depth of the findings results in discussion and address the comments.

Minor comments:

1. I strongly suggest that you spell out all abbreviations in the text the first time mentioned in the text.

2. English needs to be improved, for which they must get the assistance of someone with a native command of the English language. 

3. Cross-reference all of the citations in the text with the references in the reference section and make sure that all references have a corresponding citation within the text and vice versa.

L27: add crop yield losses due to the soil salinity

L52: add more information about PGPR- Burkholderia sp.

L60: explain the isolation of PGPR and selection.  

L63: primers 27F and 1492R- add sequences

L112: 16 standard monosaccharides- company?

L133: “/” revise it

L134: BKEPS?

L181: to improve the quality and clarity of the discussion part, I suggest that you provide more specific examples of previous work in the field and explain how your findings relate to and build upon that work. Additionally, it may be helpful to clearly state the specific focus or scope of the discussion.

L191: how you check the non-pathogenic (Burkholderia sp. BK01)? have you done any test? as Burkholderia gladioli having lots of reports. Provide the non-pathogenicity.

L261: add units. Write the abbreviation.

L290: enlarge the Figure 4. Elaborate in details.

L295: what is the major reason chlorophyll significantly dropped in Salt (100mM NaCl)

L296: Arabidopsis- italic

L348: revise the conclusion by providing more justification and highlighting the novelty of your work, emphasizing the practical applications of your findings.

Comments on the Quality of English Language

English needs to be improved. 

Reviewer 2 Report

Comments and Suggestions for Authors

A publication entitled 'Polysaccharides produced by PGPR strain Burkholderia sp. BK01 enhance salt stress tolerance to Arabidopsis thaliana' EPS obtained from cultures of the Burkholderia strain. In their work, the authors isolated a new bacterial isolate and, as part of the culture optimisation, obtained EPS that showed properties that enhanced salt stress tolerance in Arabidopsis thaliana. Additionally, the basic biochemical properties of the obtained isolates were determined.

The following is a review of the above work

General comments

Throughout the paper, there is a lack of space between words and citation numbers, for example, Line 31, line 35-36, line 107. The authors must carefully edit and reformat the manuscript before sending it back to the editor.

The authors described that the isolate they obtained was similar to Burkholderia gladioli. Bacteria from this species can cause diseases in humans and/or plants. Studies on the pathogenicity of this isolate are planned?

Introduction

Line 27-29 - the authors write about the big problem of soil salinisation and what effects it has on plants. However, it would be worthwhile to add 1-2 sentences on the origin of this problem. What is its source, especially in soils that are native to the authors?

The authors described the problem of salinity and the properties of PGPR bacteria in general in the introduction, but forgot to outline the properties and uses of EPS, which is the main core of the paper. In the Introduction, it is necessary to show that EPS or other natural polymers are produced by microorganisms and are widely used in many areas of life to further demonstrate where the polymers obtained by the authors are located.

Materials and Methods

Line 60: add a dot after Triticum aestivum L.

Line 63 - the full sequences of the primers used can be added

Line 73 - digit 3 in Ca3(PO4) appears not to be subscripted

Line 81 - lowercase t in the word L-tryptophan

Line 102-103 -: what proportion of ethyl alcohol is used in precipitation ?

The Line 106-107 - the authors describe that they determined the concentration of sugars and proteins in the obtained EPS. Was EPS fully soluble in water? Or in another solvent ?

Line 127 - synchronised seeds. What does this phrase mean ?

Section 2.4 - How did the authors inoculate the plants with an EPS suspension? Was it a soil inoculation ? Was a mixture of NaCl and EPS used, or after what time EPS was added ? Why did the authors choose an EPS concentration of 100ppm which is a concentration of the order of 0.01% ?

Line 141 - must be added by whom is this method. Method by ?

Line 156-157 - the mixture was centrifuged after boiling ?

Section 2.5.4. The method did not state the units in which enzyme activity was calculated. It is worth adding this already in this passage

Results and discussion

Line 193 - italics

Line 206 - Pi-solubilisation, not better just P-solubilisation

Line 249-250 - when describing the isomers of sugar monomers, use d written in capitalized form D-

Figure 3 - The authors described the viscosity of the EPS/postculture liquid in the graph and text, but did not include information in the methodology section on how these measurements were performed. This should be supplemented

Table 2 - fresh mass can be expressed in milligrams (mg)

General comments on the entire chapter. The authors have combined the description of the results and discussion. While the description of the results is correct for a research paper (in places where it would need a little expansion), the discussion section itself is written very poorly or in many parts not at all. The discussion is a comparison of our results with those of other authors. Referring to other groups of bacteria or microorganisms. Even if a mechanism is unknown, and there is a lot of research on EPS and even more on bacteria, one should show what the authors have achieved in relation to the current state of knowledge. I recommend rewriting this section to add more information or to break this chapter into two separate chapters. This is necessary in order for the work to be published

Round 2

Reviewer 1 Report

Comments and Suggestions for Authors

The manuscript has improved accordingly and thanks for attending all the suggestions. Wish you all the best.

Comments on the Quality of English Language

Minor editing required.

Reviewer 2 Report

Comments and Suggestions for Authors

The authors have addressed the comments suggested by the reviewer. As a result, the paper they presented gained in value. However, the authors must make a few more corrections before the above article can be published.

The authors have answered the questions posed to them in the attached file. However, they did not include all of these in the text. It is in the idea of reviewing scientific articles to point out deficiencies in the paper so that the potential reader can understand exactly what methodology was used and how the results were obtained. Responses to the reviewer, without specific changes to the text, add nothing to the paper.

The authors should add information:

Line 28-30 - the SOURCE of the soil salinity problem should be added

Chapter 2.2.2 - the authors have not added the original citation to the Salkowski paper suggested in the previous review

Chapter 2.3.1 - add information on alcohol ratio

Chapter 2.3.2 - add information on the solubility of the EPS obtained

Chapter 2.4 - Exact information on how EPS was added to the soil and when NaCl was introduced. This is important information

Table 1 and line 268–the notation of the structure of the sugar monomers is still incorrect. We do not use an uppercase D, nor a lowercase d, only the notation capitals (D) - this is the notation of the letter d by formatting the capitals in the word program options

Figure 3 - information on how viscosity is measured should be added TO the methodology, not in the graph description

The Results and Discussion section has been expanded with a discussion referring to other works. Several chapters could still be expanded to include more examples
